

# Reduced dynamics of full counting statistics

Felix A. Pollock[1], Emanuel Gull[2], Kavan Modi[1,*] and Guy Cohen[3,4,†]

1 School of Physics & Astronomy, Monash University, Clayton, Victoria 3800, Australia
2 Department of Physics, University of Michigan, Ann Arbor, Michigan 48109, USA
3 School of Chemistry, Tel Aviv University, Tel Aviv 69978, Israel
4 The Raymond and Beverley Sackler Center for Computational Molecular and Materials
Science, Tel Aviv University, Tel Aviv 6997801, Israel

* kavan.modi@monash.edu, † gcohen@post.tau.ac.il

## Abstract

We present a theory of modified reduced dynamics in the presence of counting fields. Reduced dynamics techniques are useful for describing open quantum systems at long emergent timescales when the memory timescales are short. However, they can be difficult to formulate for observables spanning the system and its environment, such as those characterizing transport properties. A large variety of mixed system–environment observables, as well as their statistical properties, can be evaluated by considering counting fields. Given a numerical method able to simulate the field-modified dynamics over the memory timescale, we show that the long-lived full counting statistics can be efficiently obtained from the reduced dynamics. We demonstrate the utility of the technique by computing the long-time current in the nonequilibrium Anderson impurity model from short-time Monte Carlo simulations.

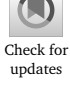
# 1 Introduction

Nonequilibrium dynamics in open quantum systems that are strongly correlated with their environment are of great interest in a variety of fields. They have long played a central role in mesoscopic transport [1, 2] and molecular electronics [3]. Experimental investigations of these systems continue to elucidate the consequences of quantum coherence in nonequilibrium situations [4–10]. More recently, they have become relevant to understanding transport phenomena in ultracold atomic systems [11–14]. Since such systems are typically driven away from equilibrium, they are characterized by the way charge and energy flow through them. A great deal of attention is therefore focused on understanding *transport properties* and their *fluctuations* or *counting statistics* [15–17].

However, the theoretical treatment of interacting nonequilibrium quantum systems remains a challenging endeavor where analytical results and reliable approximations are relatively sparse. In regimes where no reliable approximations are known to exist, numerically exact methods are needed. Such methods often proceed by starting from a well-defined, tractable initial condition and propagating it to the nonequilibrium steady state. If the exact wavefunction is used to represent the state of the system, this task requires a full solution of the many-body Schrödinger equation, and the computational complexity rises exponentially with the system's size, such that large systems cannot be studied.

One class of problems where the computational scaling in system size can often be bypassed are *quantum impurity models*. These describe systems where many body interactions are restricted to a small physical region $S$. This, in turn, is coupled to a large (or infinite) environment $E$ whose constituents are assumed to be noninteracting in the sense that their Hamiltonian is quadratic. Research on numerical techniques for simulating nonequilibrium dynamics in impurity models is a lively and active field [18]. A few examples of paradigms where significant recent advances have been made are matrix product state representations [19–25], hierarchical equation of motion techniques [26–30] and quantum Monte Carlo algorithms [31–42]. Nevertheless, in many of the most successful methods it remains computationally difficult to reliably perform propagation to long times. *I.e.*, while systems with large baths can be studied, the scaling in time is either exponential or polynomial, but in any case greater than linear. Some approaches for bypassing this issue rely on *reduced dynamics* (RD): effective non-Markovian equations of motion for quantities with the (Hilbert space) dimensionality of the small interacting region $S$ only. If the kernel representing non-Markovian effects is short-ranged in time, evaluating it up to a cutoff time provides enough information to inexpensively obtain dynamics at long and even infinite timescales [43–49].

RD provides access to the time dependence of all *local* observables, *i.e.* observables within the small interacting region. A major disadvantage of RD is that observables that are of interest in the study of transport are often *nonlocal*. While RD of nonlocal observables can be formulated [50], their use requires the evaluation of specialized, observable-dependent nonlocal quantities that can be challenging to calculate in practice.

An elegant theoretical alternative to the direct evaluation of nonlocal observables is to introduce nonlocal *counting fields* into the Hamiltonian. If dynamics with respect to the modified Hamiltonian are evaluated, it is then possible to obtain nonlocal observables, their fluctuations and all their higher moments, which are collectively termed the full counting statistics (FCS), from a local *generating function*. Calculating the FCS generating function involves the evaluation of a specialized kind of auxiliary, counting-field-modified dynamics that is nonunitary not just in the system subspace, but for the overall system + environment. Simulating such dynamics poses deep technical challenges within many methods, but several approaches exist in the literature and recent advances have been made towards computing FCS [51–60]. Among these, so far only the methods of Refs. [55–57] are able to account for FCS in models relevant

to electronic transport. Yet, existing RD techniques are not able to account for FCS as they generally assume unitarity of the overall dynamics.

In this Article, we develop a theory of RD in the presence of counting fields. We build on two major formalisms used in the literature, the Nakajima–Zwanzig–Mori equation (NZME) and its discrete-time version, known as the transfer tensor (TT) method. In Sec 2 and 3, we derive continuous and discrete memory kernel approaches to evaluating generating functions, respectively. In Sec 4, we then demonstrate our approach on the nonequilibrium Anderson impurity model, a canonical representation of interacting quantum transport.

## 2 NZME for FCS

We will present two theoretical results: first, a derivation of RD that are continuous in time; then, a derivation of RD that are stroboscopic in time (i.e. defined at a discrete set of times). While we will apply only the stroboscopic formalism to the simulations below, we begin with the continuous formalism for two reasons. First, it provides a more explicit connection to the (more fundamental) continuous time dynamics that will be most familiar to many readers; second, it opens the door to the application of other numerical techniques that may be better suited to the continuous time domain.

Master equations are often employed to describe the RD of open quantum systems. Memoryless or Markovian systems, such as electronic states in atoms weakly coupled to a wideband photonic cavity, can be described by the Gorini–Kossakowski–Sudarshan–Lindblad (GKSL) master equation, $\dot{\rho}_t^s = \mathcal{L}_t^s \rho_t^s$. This equation, which describes the dynamics of observables in the "system" subspace $S$, is celebrated for its simplicity and desirable features, such as the complete positivity of its solutions. However, many open quantum systems in condensed matter physics and quantum chemistry involve complex, non-Markovian memory effects not included in the GKSL description.

Corrections starting from the Markovian weak-coupling limit have been widely discussed in the literature [61–65]. To derive such corrections, it is useful to begin from the exact expression, which can be obtained by tracing out the environment or bath subspace. This is the Nakajima–Zwanzig–Mori equation (NZME):

$$\dot{\rho}_t^s = \mathcal{L}_t^s \rho_t^s + \int_{t_0}^{t} ds \, \mathcal{K}_{t,s}^s \rho_s^s + \mathcal{J}_{t,t_0}^s. \tag{1}$$

Here, $\mathcal{L}_t$ is a time-local Liouvillian; $\mathcal{K}_{t,s}$ is a memory kernel superoperator quantifying the effect of the baths; and $\mathcal{J}_{t,t_0}$ is an inhomogeneous term that encodes initial correlations between the $S$ subspace and the baths. Various approximate master equations emerge from the NZME at specific limits: for example, when $\mathcal{K}_{t,s}$ is nonzero only for $t = s$ and $\mathcal{J}_{t,t_0}$ vanishes we recover the GKSL master equation.

One is often interested in observables not fully contained within the system subspace $S$, such that their expectation values and higher-order fluctuations cannot be computed from $\rho_t^s$. An example commonly considered in quantum transport is energy or particle currents on the interface of the system with a particular bath. These can be expressed as the time derivatives of an observable in the full Hilbert space: in this case, the total energy or occupation in the bath, respectively. They are therefore system–bath or bath–bath observables. A convenient and formally exact way to treat such observables, along with all their higher-order moments and cumulants, is by introducing a counting field [15]. For any system, bath, or system–bath observable $A_t$—that is a driven observable in the Schrödinger picture such that it commutes with the initial density matrix, i.e., $[A_{t_0}, \rho_{t_0}^{SE}] = 0$—we can compute the $m$th moment by

differentiating the generating function $Z_{\lambda,t}$:

$$\langle(\Delta A)_t^m\rangle = \lim_{\lambda\to 0}\frac{\partial^m}{\partial(i\lambda)^m}Z_{\lambda,t} = \sum_{k=0}^m \binom{m}{k}\langle\tilde{A}_t^{m-k}A_{t_0}^k\rangle \tag{2}$$

$$= \sum_{k=0}^m (-1)^k\binom{m}{k}\operatorname{Tr}\left\{A_t^{m-k}U_{t:t_0}A_{t_0}^k\rho_{t_0}^{SE}U_{t:t_0}^\dagger\right\}. \tag{3}$$

Here, $\lambda$ is the counting field, $\tilde{A}_t$ is the Heisenberg-picture version of $A_t$, and $Z_{\lambda,t} = \operatorname{Tr}[\zeta_{\lambda,t}^s]$, where $\zeta_{\lambda,t}^s$ is a generalised density operator for the system that can be expressed in terms of the joint $S\otimes E$ dynamics as

$$\zeta_{\lambda,t}^s \equiv \operatorname{Tr}_E\left\{e^{\frac{i\lambda A_t}{2}}U_{t:t_0}e^{\frac{-i\lambda A_{t_0}}{2}}\rho_{t_0}^{SE}e^{\frac{-i\lambda A_{t_0}}{2}}U_{t:t_0}^\dagger e^{\frac{i\lambda A_t}{2}}\right\}, \tag{4}$$

with $U_{t:s}$ the usual (unitary) time evolution operator.

This can be expressed more concisely as $\zeta_{\lambda,t}^s = \operatorname{Tr}_E\{\mathcal{Z}_{\lambda,t:t_0}(\rho_{t_0}^{SE})\}$, in terms of the $\lambda$-dependent time-evolution superoperator

$$\mathcal{Z}_{\lambda,t:s} = \mathcal{A}_{\lambda,t}^+\circ\mathcal{U}_{t:s}\circ\mathcal{A}_{-\lambda,s}^+, \tag{5}$$

where $\mathcal{A}_{\lambda,t}^\pm(X) = e^{i\lambda A_t/2}Xe^{\pm i\lambda A_t/2}$ and $\mathcal{U}_{t:s}(X) = U_{t:s}XU_{t:s}^\dagger$. Here, $\circ$ denotes composition of superoperators. Finally, note that the $m$th cumulant $C_t^m$ can be computed from the logarithm of the generating function as $C_t^m = \lim_{\lambda\to 0}\frac{\partial^m}{\partial(i\lambda)^m}\ln(Z_{\lambda,t})$.

We now come to our first result: the construction of an exact memory kernel equation of motion for the operator $\zeta_{\lambda,t}$ with fixed $\lambda$, and hence for the generating function $Z_{\lambda,t}$. We will also show that this reduces to the NZME in the case where $\lambda = 0$. The principal effect of a finite $\lambda$ with respect to the normal dynamics is that the generator will no longer be a simple commutator with the $S\otimes E$ Hamiltonian. The time derivative of the modified density matrix can, instead, be written as

$$\partial_t\mathcal{Z}_{\lambda,t:s} = \mathcal{R}_{\lambda,t}(\mathcal{Z}_{\lambda,t:s}), \tag{6}$$

where

$$i\,\mathcal{R}_{\lambda,t}(X) = \mathcal{A}_{\lambda,t}^-(H_t)X - X\mathcal{A}_{-\lambda,t}^-(H_t) - \frac{1}{2}\int_0^\lambda d\tau\left\{\mathcal{A}_{\tau,t}^-(\partial_t A_t)X + X\mathcal{A}_{-\tau,t}^-(\partial_t A_t)\right\}. \tag{7}$$

For fixed $\lambda$, the evolution superoperator is divisible:

$$\mathcal{Z}_{\lambda,t:t_0} = \mathcal{Z}_{\lambda,t:s}\mathcal{Z}_{\lambda,s:t_0}, \quad\text{for}\quad t > s > t_0. \tag{8}$$

We can therefore follow the steps of the standard derivation of the NZME (see e.g. Refs. [66, 67] and Appendix A) to write

$$\partial_t\zeta_{\lambda,t}^s = \mathcal{S}_{\lambda,t}^s\zeta_{\lambda,t}^s + \int_{t_0}^t ds\,\mathcal{M}_{\lambda,t,s}^s\zeta_{\lambda,s}^s + \mathcal{N}_{\lambda,t,t_0}^s, \tag{9}$$

with $\mathcal{S}_{\lambda,t}^s$, $\mathcal{M}_{\lambda,t,s}^s$, and $\mathcal{N}_{\lambda,t,t_0}^s$ the finite $\lambda$ counterparts to $\mathcal{L}_t^s$, $\mathcal{K}_{t,s}^s$ and $\mathcal{J}_{t,t_0}^s$ in Eq. (1). These can be expressed in terms of $S\otimes E$ projection superoperators $\mathcal{P}$ and $\mathcal{Q} = \mathcal{I} - \mathcal{P}$, with action $\mathcal{P}(X) = \operatorname{Tr}_E[X\otimes\sigma^E]$ a function of reference state $\sigma^E$, as

$$\mathcal{S}_{\lambda,t}^s X^s = \operatorname{Tr}_E\left\{\mathcal{P}\mathcal{R}_{\lambda,t}\mathcal{P}(X^s)\right\},$$
$$\mathcal{M}_{\lambda,t,s}^s X^s = \operatorname{Tr}_E\left\{\mathcal{P}\mathcal{R}_{\lambda,t}\mathcal{G}_{\lambda,t,s}\mathcal{Q}\mathcal{R}_{\lambda,t}\mathcal{P}(X^s)\right\}, \tag{10}$$
$$\mathcal{N}_{\lambda,t,t_0}^s = \operatorname{Tr}_E\left\{\mathcal{P}\mathcal{R}_{\lambda,t}\mathcal{G}_{\lambda,t,s}\mathcal{Q}\rho_{t_0}^{SE}\right\},$$

where

$$\mathcal{G}_{\lambda,t,s} = \mathrm{T}\exp\left[\int_s^t dr\, \mathcal{Q}\mathcal{R}_{\lambda,r}\right],\tag{11}$$

with T a time ordering operator.

Since $\mathcal{A}_{0,t}^{\pm} = \mathcal{I}$, it follows from Eq. (7) that $\mathcal{R}_{0,t}$ is the usual generator of the von Neumann equation, i.e., $\mathcal{R}_{0,t}(X) = -i[H_t, X]$. Therefore, in the case that $\lambda = 0$, Eq. (9) reduces to Eq. (1) with $\mathcal{L}_t^s = \mathcal{S}_{0,t}^s$, $\mathcal{K}_{t,s}^s = \mathcal{M}_{0,t,s}^s$ and $\mathcal{J}_{t,t_0}^s = \mathcal{N}_{0,t,t_0}^s$ the quantities appearing in the usual NZ master equation.

As is the case without a counting field, $\mathcal{N}_{\lambda,t,t_0}^s$ goes to zero and the equation simplifies when the initial condition factorises as $\rho_{t_0}^{SE} = \rho_{t_0}^s \otimes \rho_{t_0}^E$ and the reference state is chosen as $\sigma^E = \rho_{t_0}^E$. The complexity of propagating the system to long times can still be high when there is a nontrivial memory kernel $\mathcal{M}_{\lambda,t,s}^s$ [43–45]. There exist techniques for expanding the propagator in Eq. (11), such that a closed form expression for the memory kernel in terms of system quantities can be found [43]. Therefore, in practice, one can often fit the memory kernel to data from short time numerical simulations and extrapolate to longer times under the assumption that there is a time $t_m$, beyond which memory effects are small [44–47,49,68–70].

However, numerical (or experimental) data is typically recorded on a discrete time lattice, whose spacing may be longer than the shortest dynamical timescale for the system, precluding an accurate smooth reconstruction of the memory kernel. This motivates the discrete-time transfer tensor approach [71, 72], which we now show also generalizes to the setting of full counting statistics.

## 3 Transfer tensor approach

The primary object with which we will be dealing within the TT approach is the generalized dynamical map, defined as

$$\Lambda_{\lambda,t:s}(X) = \mathrm{Tr}_E\left\{\mathcal{Z}_{\lambda,t:s}(X \otimes \sigma^E)\right\}.\tag{12}$$

For a factorising initial condition with $\sigma^E = \rho_{t_0}^E$, we have $\zeta_{\lambda,t}^s = \Lambda_{\lambda,t:t_0}(\rho_{t_0}^s)$. As before, when $\lambda = 0$ these become the dynamical maps of the usual time evolution, and must be completely positive and trace preserving. More generally, however, they need not take physical density operators to physical density operators.

In the following two subsections, we will show how the approach is formulated and applied. The first of these outlines the procedure for extracting dynamical maps from short-time values for the generalized density operator $\zeta_{\lambda,t}^s$. The second shows how a family of transfer tensors can be extracted from these dynamical maps, and how the former can be used to approximate long-time data.

### Constructing the dynamical maps from data

In principle, maps $\Lambda_{\lambda,t:s}$ can be constructed from experimental or simulated data with sufficiently many observations and initial conditions using standard ideas from quantum process tomography [73, 74]. Namely, we need the scalars associated with

$$\Lambda_{\lambda,t:s;\beta\alpha} \equiv \mathrm{Tr}\left[Y_\beta \Lambda_{\lambda,t:s}(X_\alpha)\right],\tag{13}$$

where $X_\alpha$ and $Y_\beta$ are input and output operators, respectively. In practice, only approximate data points $L_{\lambda,t:s;\beta\alpha} \simeq \Lambda_{\lambda,t:s;\beta\alpha}$ are available (from simulation or experiment). With access

to estimates of the scalar coefficients corresponding to a basis set of operators $\{X_\alpha\}$ and $\{Y_\alpha\}$ on the input and the output spaces, respectively, we can fully determine the map. The only requirements are that each set should linearly span the full system operator space with $d^2$ linearly independent operators for a $d$-dimensional Hilbert space, and that the elements (in product with $\mathbb{I}^E$) should commute with $A_{t_0}$.

For any set of basis matrices there exists a dual set, that we denote as $\{\check{X}_\alpha\}$ and $\{\check{Y}_\alpha\}$, satisfying $\mathrm{Tr}[\check{W}_\alpha^\dagger W_\beta] = \delta_{\alpha\beta}$ with $W \in \{X, Y\}$. Given this, the superoperator in the Liouville form or "$A$-form" is written as

$$\Lambda_{\lambda, t:t_0} \simeq \sum_{\alpha,\beta} L_{\lambda, t:t_0; \beta\alpha} |\check{Y}_\beta\rangle\!\rangle\langle\!\langle\check{X}_\alpha|, \tag{14}$$

with $|W\rangle\!\rangle$ indicating vectorisation of the operator $W$. The last equation is self-consistent due to the fact that $\langle\!\langle\check{W}_\alpha|W_\beta\rangle\!\rangle = \mathrm{Tr}[\check{W}_\alpha^\dagger W_\beta] = \delta_{\alpha\beta}$.

In an experiment, the sets $\{X_\alpha\}$ and $\{Y_\alpha\}$ could represent sets of initial condition density operators and observables, respectively. However, when using simulated data, it is convenient to pick the self-dual basis $\{X_\alpha\} = \{Y_\alpha\} = \{|\mu\rangle\langle\nu|\}$ for some orthonormal basis of vectors $\{|\mu\rangle\}$ in the system Hilbert space. In this case, Eq. (14) reduces to

$$\Lambda_{\lambda, t:t_0} \simeq \sum_{\mu'\nu',\mu\nu} L_{\lambda, t:t_0; \mu'\nu'\mu\nu} |\mu'\rangle|\nu'\rangle\langle\mu|\langle\nu| . \tag{15}$$

The data points $L_{\lambda, t:t_0; \mu'\nu'\mu\nu}$ approximate the $\mu'\nu'$ matrix elements $\langle\mu'|\Lambda_{\lambda, t:s}(|\mu\rangle\langle\nu|)|\nu'\rangle$ of the evolved generalised density operator corresponding to an input density matrix with a single non-zero matrix element at entry $\mu\nu$.

As noted earlier, when $\lambda = 0$ the dynamical maps must be completely positive and trace preserving (provided there are no initial system–environment correlations). These conditions are sufficient to ensure that, even when applied to a subsystem of a larger composite, physical density operators are mapped to physical density operators. Complete positivity can be most easily expressed in terms of the so-called Choi form of the map: specifically, using the same notation as Eq. (14), the operator $\sum_{\alpha,\beta} L_{\lambda, t:t_0; \beta\alpha} \check{Y}_\beta^\dagger \otimes \check{X}_\alpha^*$ should be positive semi-definite. Trace preservation, i.e., $\mathrm{Tr}[\Lambda_{\lambda, t:s}(X)] = \mathrm{Tr}[X]$ can be expressed in terms of the superoperator matrix form of the map as $\langle\!\langle\mathbb{I}| \Lambda_{0, t:t_0} = \langle\!\langle\mathbb{I}|$. When starting from imperfect data, enforcing complete positivity and trace preservation numerically can greatly reduce the potential for unphysical propagated dynamics. Beyond simple projection techniques, there exist several more sophisticated methods for reconstructing physical dynamical maps from incomplete or noisy data [75,76].

For finite values of $\lambda$, there are no analogous universal conditions that the dynamical maps must satisfy. However, we note in passing that by expanding Eqs. (5) and (12) in powers of $\lambda$, one finds that for small $\lambda$ the counterpart of the trace preservation condition can be expressed as

$$\langle\!\langle\mathbb{I}| \Lambda_{\lambda, t:t_0} = \langle\!\langle\mathbb{I}| + i\lambda\left(\langle\!\langle\tilde{A}_t^s| - \langle\!\langle\tilde{A}_0^s|\right) + O(\lambda^2). \tag{16}$$

Here, $\tilde{A}_t^s = \mathrm{Tr}_E[U_{t:0}^\dagger A_t U_{t:0}\mathbb{I} \otimes \sigma_E]$ is the Heisenberg picture counting operator averaged with respect to the environment state. If this operator can be estimated independently, then numerical enforcement of Eq. (16) could be used to stabilize the construction of the dynamical maps. It is also possible that detailed balance conditions [77] stemming from fluctuation theorems [78] may be generally useful in this regard, though this remains speculative at present.

## Propagation using transfer tensors

We now show how the short-time dynamical maps can be used to extrapolate the dynamics to arbitrarily long times using the TT method. Following the same steps as in Ref. [67] (re-

produced in Appendix B), one can recursively define a family of transfer tensors on a set of discrete times $\{t_j\}$ through

$$T^{(1)}_{\lambda,t_j:t_{j-1}} = \Lambda_{\lambda,t_j:t_{j-1}} \quad \text{and} \quad T^{(n)}_{\lambda,t_j:t_{j-n}} = \Lambda_{\lambda,t_j:t_{j-n}} - \sum_{k=1}^{n-1} T^{(k)}_{\lambda,t_j:t_{j-k}} \Lambda_{\lambda,t_{j-k}:t_{j-n}}, \tag{17}$$

such that dynamical maps between distant times can be expressed in terms of intermediate ones as

$$\Lambda_{\lambda,t_j:t_{j-n}} = \sum_{k=1}^{n} T^{(k)}_{\lambda,t_j:t_{j-k}} \Lambda_{\lambda,t_{j-k}:t_{j-n}}, \tag{18}$$

with $\Lambda_{\lambda,t_{j-n}:t_{j-n}} = \mathcal{I}$. In the case that the $S \otimes E$ Hamiltonian $H_t$ and the counting operator $A_t$ are time-independent, the generalised dynamical maps become stationary; that is, $\Lambda_{\lambda,t_j:t_k} = \Lambda_{\lambda,t_j-t_k:0} \equiv \Lambda_{\lambda,j-k}$. Hence, for an evenly spaced time-grid with $t_j - t_{j-1} = \delta t$, we have $T^{(n)}_{\lambda,t_j:t_{j-n}} = T^{(n)}_{\lambda,n\delta t:0} \equiv T_{\lambda,n}$. Eqs. (17) and (18) then become

$$T_{\lambda,1} = \Lambda_{\lambda,1}, \qquad T_{\lambda,n} = \Lambda_{\lambda,n} - \sum_{k=1}^{n-1} T_{\lambda,k} \Lambda_{\lambda,n-k}, \tag{19}$$

and

$$\Lambda_{\lambda,n} = \sum_{k=1}^{n} T_{\lambda,k} \Lambda_{\lambda,n-k} \Rightarrow \zeta_{\lambda,t_n} = \sum_{k=1}^{n} T_{\lambda,k} \zeta_{\lambda,t_{n-k}}. \tag{20}$$

This equation is exact, and reminiscent of convolution with the memory kernel; the direct correspondence between the two is discussed in Refs. [67, 72] for the $\lambda = 0$ case. However, for systems with a decaying memory, the generalized density operator can be approximated by Eq. (20) with the sum truncated at some large $m < n$:

$$\zeta_{\lambda,t_n} \simeq \sum_{k=1}^{m} T_{\lambda,k} \zeta_{\lambda,t_{n-k}}. \tag{21}$$

As such, the dynamics up to time $t_m$ can be used to propagate the generalized density operator, and hence the generating function, to longer (or potentially infinite) times.

Since we typically find the propagated dynamics to be sensitive to high frequency noise in the original maps $\{\Lambda_{\lambda,n} : n < m\}$, we instead use a (symmetric) rolling average of the latter:

$$\tilde{\Lambda}_{\lambda,n} = \frac{1}{w} \sum_{k=n-w}^{n+w} \Lambda_{\lambda,n}, \qquad w = \min(N, n, m-n), \tag{22}$$

where $N$ is a parameter that sets the maximum size of the averaging window. See Ref. [79] for an alternative memory-cutoff scheme that may also be useful in this context.

# 4 Model

To demonstrate how the transfer method can be used to obtain full counting statistics in practice, we apply it to the nonequilibrium Anderson impurity model, which is often used

to describe electronic transport through junctions. This is defined by the Hamiltonian $H = H_{\mathcal{S}} + H_{\mathcal{E}} + H_{\mathcal{SE}}$, where

$$
\begin{aligned}
H_{\mathcal{S}} &= \sum_{\sigma} \varepsilon a_{\sigma}^{\dagger} a_{\sigma} + U a_{\uparrow}^{\dagger} a_{\uparrow} a_{\downarrow}^{\dagger} a_{\downarrow}, \\
H_{\mathcal{E}} &= \sum_{k\sigma} \varepsilon_k a_{k\sigma}^{\dagger} a_{k\sigma}, \\
H_{\mathcal{SE}} &= \sum_{k\sigma} v_k a_{k\sigma}^{\dagger} a_{\sigma} + v_k^* a_{\sigma}^{\dagger} a_{k\sigma}.
\end{aligned}
\tag{23}
$$

Here, $\varepsilon$ is a single-particle occupation energy on the system/impurity site, and $U$ is an on-site Hubbard interaction; the $\varepsilon_k$ and $v_k$, respectively, are occupation energies in the environment levels $k$ and system–environment coupling terms; $a_{\sigma}$ are system fermionic annihilation operators with spin $\sigma \in \{\uparrow, \downarrow\}$; and the $a_{\sigma k}$ are fermionic annihilation operators in the environment. At the continuum limit, the $v_k$ are determined by the coupling strength function $\Gamma(\omega) \equiv \sum_k |v_k|^2 \delta(\omega - \omega_k)$, which we set to

$$
\Gamma(\omega) = \frac{\Gamma}{\left(1 + e^{\nu(\omega - \Omega_C)}\right)\left(1 + e^{-\nu(\omega + \Omega_C)}\right)}.
\tag{24}
$$

This describes an environment with a flat density of states having a bandwidth $2\Omega_C$ and a band cutoff width of $1/\nu$. Throughout this work, $\Omega_C = 10\Gamma$, $\nu = 10/\Gamma$, $U = 5\Gamma$ and $\epsilon = -U/2$. The environment levels are equally distributed between a "left" and "right" subspace, denoted as $\mathcal{L}$ and $\mathcal{R}$, respectively. At the initial time the system is prepared in a factorized state $\rho = \rho_{\mathcal{S}} \otimes \rho_{\mathcal{E}}$, with the environment part further factorizing as $\rho_{\mathcal{E}} = \rho_{\mathcal{L}} \otimes \rho_{\mathcal{R}}$ and each of the two environment subspaces initially in thermal equilibrium at a given temperature $T$ and chemical potential $\mu_{\mathcal{L}/\mathcal{R}}$. By setting $\mu_{\mathcal{L}/\mathcal{R}} = \pm V/2$, a bias voltage $V$ can be applied across the system, driving it towards a nonequilibrium steady state exhibiting a nonzero current $I$.

An important and well-known physical feature of the Anderson model is the emergence of strong correlation physics at temperatures below the Kondo scale $T_K$ [80–82], which for our choice of parameters can be estimated as $\sim 0.3\Gamma$ [81]. The behavior of the NZME memory kernel for *normal dynamics* in this model was investigated in Ref. [44], showing that Kondo physics is associated with the emergence of a long memory timescale. When this timescale is still small enough to allow numerical evaluation up to it, it was also shown that dynamics up to long or infinite timescales could be obtained [44, 45].

Here, following Refs. [55] and (for the noninteracting case) [54], we will consider *modified dynamics* in the presence of a counting field $\lambda$ that generates moments of the total population in the left subspace $\mathcal{L}$. The time derivative of this population is a commonly used definition for the electronic current $I$, and the one used here.

## 5 Results

We evaluate the generalized density operator $\zeta_{\lambda, t_n}$ using a numerically exact quantum Monte Carlo technique—called inchworm Monte Carlo—that was described and validated in previous publications [55–57]. Simulations are performed for a complete set of initial conditions so that the transfer tensor can be fully recovered from Eqs. (15) and (17).

We begin by considering the transfer tensors themselves. In Figure 1 we present their $\mathbf{L}^2$ norm $\|T\|$—a rough measure of the contribution of memory effects at the corresponding time scale—for two voltages $V \in \{0, 5\Gamma\}$; at two inverse temperatures respectively above and below the Kondo regime, $\beta\Gamma \in \{0.1, 10\}$; and for three values of the counting field, $\lambda \in \{0, 0.3, 0.6\}$.

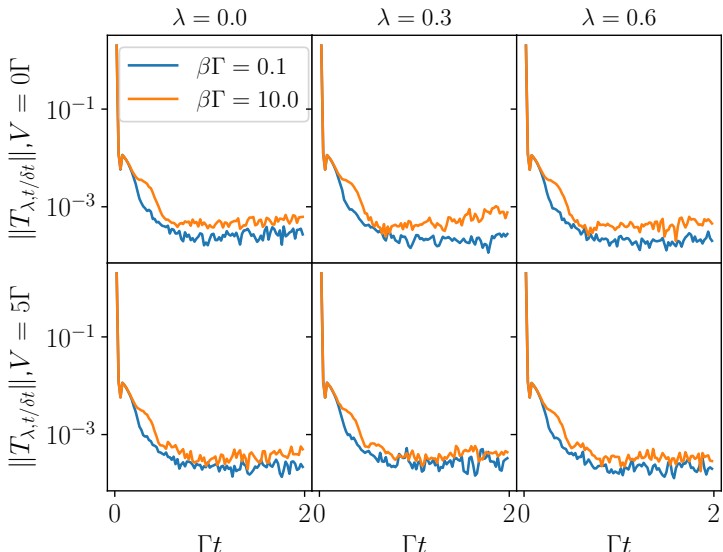

Figure 1: Norms of transfer tensors $T_{\lambda,n}$, with $n = t/\delta t$ and $\Gamma\delta t = 0.02$, as a function of the time $t$. Two bias voltages are shown: $V = 0$ (top row) and $V = 5\Gamma$ (bottom row). Also plotted are several values of the counting fields, $\lambda = 0.0$ (left column), $\lambda = 0.3$ (middle column), and $\lambda = 0.6$ (right column); and two inverse temperatures $\beta\Gamma = 0.1$ (blue) and $\beta\Gamma = 10$ (orange).

At higher temperatures and voltages, we expect the dynamics to express only timescales commensurate with the bare energy scales in the problem. Within the Kondo regime, on the other hand, we expect timescales proportional to $T_K$ to emerge. Here, we only consider two temperatures and do not seek to explicitly reproduce the Kondo scaling, which would also require cleaner data. However, it is clear that longer memory timescales emerge at low temperature in all cases. It can also be seen that this effect is somewhat more pronounced at zero voltage, consistent with the idea that a nonequilibrium drive should suppress Kondo physics (though see Refs. [83, 84] for some qualifications to this statement).

Interestingly, Figure 1 shows that the introduction of the counting field does not strongly affect the norm of the transfer tensor. Individual elements do vary, and we will see immediately that the dynamics is dramatically modified by the field. Nevertheless, the weak response in the norm suggests that the effective range of the memory kernel depends on the counting field only weakly, at least in this regime, implying that memory cutoffs used for regular time dynamics should be equally valid at finite $\lambda$.

When the transfer tensor decays to zero rapidly enough, approximate long-time dynamics can be obtained from Eq. (21). In Figure 2 we show how this works in practice for both real (top panels) and imaginary (bottom panels) parts of the counting field $Z$. We selected the parameters used in Figure 1 that turn out to be the most difficult for this purpose: $\beta = 10/\Gamma$ and $V = 5\Gamma$. The values of the counting field are the same as those used in Figure 1. Numerically exact benchmarks from inchworm Monte Carlo are shown in black dashed lines up to time $\Gamma t = 8$, indicated by the lightly shaded region. Reconstructed dynamics is shown in red up to twice that timescale, and can be extended in time at negligible numerical cost.

Going from left to right, in each column of panels the cutoff time $t_m$ (dashed vertical line) is increased. Only data in the darkly shaded regions to the left of $t_m$ is used in constructing the transfer tensors. In the regime we considered, one can see that a cutoff time of $t_m = 1.2/\Gamma$ (right panels) is sufficient for reproducing the dynamics at a reasonable level of accuracy. We note that within our present implementation of the method, increasing the memory time does

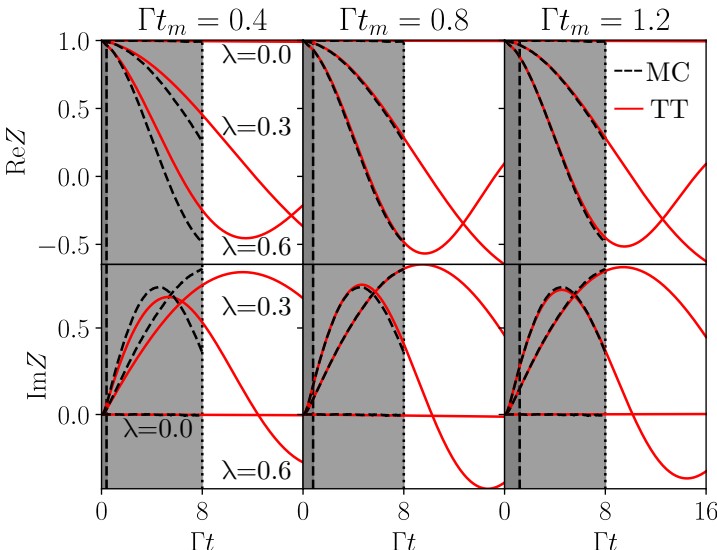

Figure 2: Reconstructed generating functions $Z_{\lambda,t}$ (red, solid) at memory cutoff times $\Gamma t_m = 0.4$ (left column), $\Gamma t_m = 0.8$ (middle column), and $\Gamma t_m = 1.2$ (right column). The time interval up to the cutoff time, which is used to construct the transfer tensor, is shaded in dark gray. The values of the counting fields are as in Figure 1, and can be distinguished by respectively increasing curvature. The temperature is set to $\beta = 10/\Gamma$, the voltage to $V = 5\Gamma$, and the system is initially in the unoccupied state. A smoothing parameter $N = 6$ was used. As a benchmark, full inchworm Monte Carlo simulations are also shown (black, dashed) within the light gray region.

not always improve the accuracy. This, because going to longer times typically also increases the noise; the degree to which this occurs depends on the underlying numerical method for evaluating the modified dynamics. The reconstructed dynamics are especially sensitive to noise near the cutoff time, though the smoothing parameter provides some robustness. We will demonstrate this and remark further on it below, where the effect is more obvious.

Next, we consider the dynamics of the current $I(t)$ obtained from the first cumulant of the generating function. We will be particularly interested in the effect of noise and smoothing on the accuracy of the reconstruction. Therefore, in Figure 3, we show relatively noisy numerical data from inchworm Monte Carlo in black. Two initial conditions are shown in the two panels. Reconstructed dynamics at three different cutoff times $\Gamma t_m \in \{0.3, 0.8, 1.2\}$ are shown as solid blue, orange and green curves, respectively. For each cutoff time, we also show the effect of smoothing the input data with smoothing parameter $N = 6$ as dashed curves in the same colors.

For the unoccupied initial condition (top panel in Figure 3), the current rises rapidly, then drops to a plateau at a lower value. This is easily understood when considering that our definition of the current is the time derivative of the population in the left lead, commonly referred to in the literature as the "left" current $I_L$. When the dot is initially empty, it is rapidly filled by the left lead at short times at timescales of order $1/\Gamma$, until it reaches its steady-state population—at the parameters used here, a half-occupied state. At that point, transport is partially blocked, and the steady state current is eventually reached. A complementary "right" current $I_R$ could be calculated by considering the time derivative of populations in the right lead. The right current is suppressed at short times for this initial condition, and the "average" current $(I_L + I_R)/2$ therefore does not feature the sudden rise and fall at short times.

On the other hand, in the magnetized initial condition, transport from the left lead into

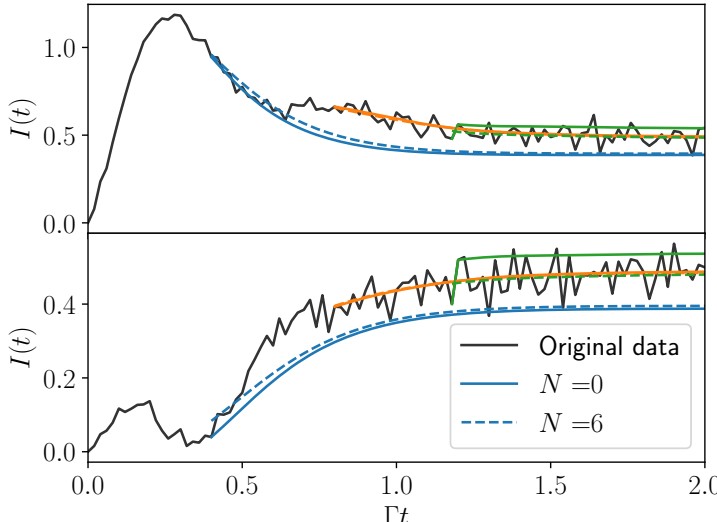

Figure 3: Reconstructed time-dependent current $I(t)$ for memory cutoff times $t_m = 0.4$ (blue), $t_m = 0.8$ (orange), and $t_m = 1.2$ (green), with (dashed) and without (solid) averaging of initial data with maximum window radius $N = 6$. For comparison, current calculated from the inchworm Monte Carlo data used in the reconstruction is shown in black. The top panel shows the current starting from the unoccupied initial condition, while the bottom panel shows that starting from a magnetized initial condition. Model parameters: $U = 5\Gamma$, $V = 5\Gamma$, $\beta\Gamma = 1.0$, $\epsilon = 0.0$, $\omega_c = 10.0\Gamma$.

the impurity is suppressed at very short times for one spin by Pauli exclusion, and for the other by the interaction. However, the initial electron on the dot rapidly empties into the right lead, and eventually the same steady state is reached.

When the cutoff time is too short (blue curves), a systematic error in the dynamics is observed. At long enough cutoff times (orange and green curves), the dynamics converges to within the noise of the Monte Carlo data. However, the unsmoothed green lines at the largest cutoff time are shifted away from the orange line by a stochastic fluctuation in both cases. This shifting survives to long times and produces a systematic error in the current. Smoothing largely rectifies this issue—the dashed orange and green curves are essentially indistinguishable from each other—but may also introduce a systematic bias that degrades accuracy. On the other hand, smoothing has negligible effect at the intermediate cutoff time (orange curves) and a small effect at short cutoff time (blue curves).

We note that due to the high peak at short times for the unoccupied initial condition, the scale of the top panel is higher. This makes it appear as if the effects of noise are different, but they are actually quite similar in practice.

The results clearly demonstrate that it is possible to reproduce dynamics at intermediate timescales with our method. However, they also show that the method is very sensitive to both cutoff time and noise. Therefore, we expect that more robust schemes for constructing the transfer tensor will be needed to enable high-precision dynamical applications.

Finally, we present results for the steady state current $\langle I \rangle_{ss} \equiv \lim_{t\to\infty} \langle I(t) \rangle$ in Figure 4. Steady state properties are often difficult to extract from methods based on time propagation, because it can be expensive to obtain accurate data sufficiently close to the long time limit. We consider the interacting, driven case $U = 5\Gamma$ and $V = 5\Gamma$, at high and low temperatures ($\beta\Gamma = 0.1, 10$). Inchworm Monte Carlo calculations are performed up to time $\Gamma t = 8$. We take the mean and standard error of the current at times $\Gamma t > 7$ as a benchmark

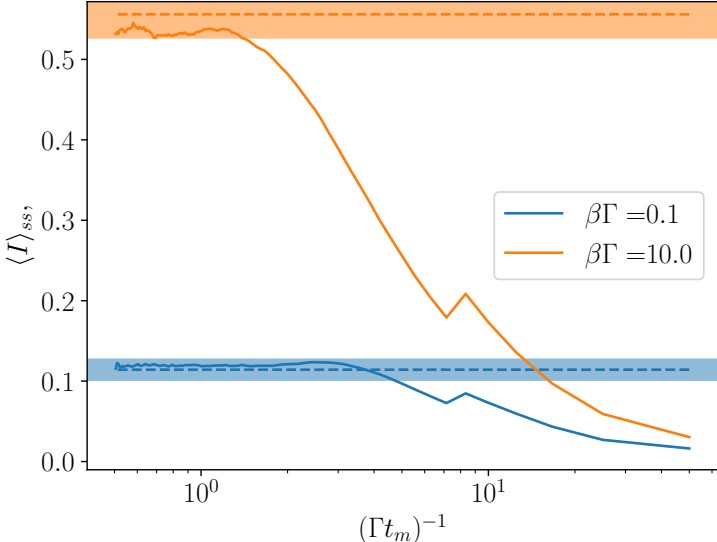

Figure 4: Reconstructed steady state current as a function of memory cutoff at inverse temperatures $\beta\Gamma = 0.1$ (blue) and $\beta\Gamma = 10.0$ (orange). All values were computed by averaging the propagated current at long times with a time step $\mathrm{d}t = 0.02/\Gamma$ and averaging parameter $N = 6$. Model parameters: $\epsilon = -2.5\Gamma$, $\omega_c = 10.0\Gamma$. Shaded regions indicate a 68% confidence interval of the steady state current reconstructed from long time Monte Carlo data (mean shown as dashed line).

estimate for the steady state value (shown as horizontal dashed lines, with shaded regions indicating the standard error). Steady state currents are then extracted from the long-time limit of the TTs corresponding to this data, but truncated at different times. The results are plotted as a function of the inverse cutoff time $(\Gamma t_m)^{-1}$. At the low temperature, an accurate plateau value is reached at times $\sim 1/\Gamma$, and for the higher temperature this occurs even sooner.

The data shown in Figure 4 indicates that by using the FCS-enabled TT method presented here, steady state currents can be reliably obtained from inchworm Monte Carlo simulations up to times $\Gamma t \sim 1$ rather than $\Gamma t \sim 8$ at the parameters considered here. Specifically, the values obtained from the transfer tensor method with simulations up to those times agrees up to standard error with the values obtained from long-time Monte Carlo simulations (corresponding to the shaded regions in the figure). Since the computational cost of the Monte Carlo calculations scales quadratically with time, this represents a practical reduction factor of $\sim 64$.

## 6 Conclusions

The reduced dynamics of a system within an environment can be exactly expressed in terms of a non-Markovian memory kernel, or on a discrete time grid as a transfer tensor. When the memory is short-ranged, short-time simulations of the kernel or transfer tensor can be used to obtain long-time dynamics within the system. However, environment observables or mixed system–environment observables are not straightforward to access in this manner; treating each such observable requires the derivation of a specialized, ad-hoc reduced dynamics that can be difficult to implement in practice. On the other hand, many such observables—as well as their higher-order moments—can be expressed in terms of their full counting statistics,

which is extracted from a generating function governed by modified, nonunitary dynamics.

We presented a generalization of the transfer tensor formalism (as well as its continuous form, the Nakajima–Zwanzig–Mori equation) to full counting statistics. Using the nonequilibrium Anderson impurity model with a particle number counting field as a test case, we showed that the memory can still be short-ranged, even though the dynamics are typically characterized by long-lived oscillations. Finally, we showed that the dynamics and steady state of the generating function and its low moments (in this case, the particle current) can be reproduced from short-time numerical simulations. The reproduction is sensitive to noise in the simulation data, and future work will be needed to make it more robust. Nevertheless, we showed that in some cases the method can be paired with modern Monte Carlo algorithms, enabling the evaluation of accurate steady states at a fraction of the direct computational cost. For other simulation methods, where the barrier to long times can be exponential and stochastic noise is less prominent, our generalized transfer tensor technique may enable calculations that would otherwise be completely intractable.

## Acknowledgements

**Funding information**    KM is supported through Australian Research Council Future Fellowship FT160100073, Discovery Project DP210100597, and the International Quantum U Tech Accelerator award by the US Air Force Research Laboratory. G.C. acknowledges support by the Israel Science Foundation (Grants No. 2902/21 and 218/19) and by the PAZY foundation (Grant No. 308/19). We are grateful for the support from Australian Friends of Tel Aviv University-Monash University. EG is supported by the U.S. Department of Energy, Office of Science, Office of Advanced Scientific Computing Research and Office of Basic Energy Sciences, Scientific Discovery through Advanced Computing (SciDAC) program under Award Number(s) DE-SC0022088.

## A   Derivation of the NZM equation for generalized density operators

For any two-parameter family of (super)operators $\mathcal{X}_{t:s}$ which satisfies an equation of the form (cf. Eq. (6))

$$\partial_t \mathcal{X}_{t:s} = \mathcal{Y}_t \mathcal{X}_{t:s}, \tag{25}$$

one can derive an equation of motion in terms of a memory kernel using the Nakajima-Zwanzig-Mori projection technique, as we will now briefly show. Variations of this derivation can be found, for example, in Refs. [66, 67].

To start with, define a family of projection superoperators $\mathcal{P}_t$, with action $\mathcal{P}_t X^{SE} = \mathrm{Tr}_E\{X^{SE}\} \otimes \sigma_t^E$, with $\sigma_t^E$ a unit-trace, time-dependent environment operator, and their complement $\mathcal{Q}_t = \mathcal{I} - \mathcal{P}_t$. These obey the following identities:

$$\mathcal{P}_t \mathcal{P}_s = \mathcal{P}_t, \tag{26}$$

$$\mathcal{Q}_t \mathcal{Q}_s = \mathcal{Q}_s, \tag{27}$$

$$\mathcal{Q}_t \mathcal{P}_s = \mathcal{P}_s - \mathcal{P}_t, \tag{28}$$

$$\mathcal{P}_t \mathcal{Q}_s = 0. \tag{29}$$

Using Eq. (25) and the identity $\mathcal{I} = \mathcal{P}_t + \mathcal{Q}_t$, the dynamics of the 'relevant' and 'irrelevant'

projections of the evolution superoperator $\mathcal{X}_{t:s}$ can be shown to satisfy

$$\partial_t \mathcal{P}_t \mathcal{X}_{t:s} = \left( \mathcal{P}_t \mathcal{Y}_t \mathcal{P}_t + \dot{\mathcal{P}}_t \right) \mathcal{X}_{t:s} + \mathcal{P}_t \mathcal{Y}_t \mathcal{Q}_t \mathcal{X}_{t:s}, \tag{30}$$

and

$$\partial_t \mathcal{Q}_t \mathcal{X}_{t:s} = \left( \mathcal{Q}_t \mathcal{Y}_t \mathcal{P}_t - \dot{\mathcal{P}}_t \right) \mathcal{X}_{t:s} + \mathcal{Q}_t \mathcal{Y}_t \mathcal{Q}_t \mathcal{X}_{t:s}, \tag{31}$$

with $\dot{\mathcal{P}}_t = \partial_t \mathcal{P}_t$ satisfying $\mathcal{P}_t \dot{\mathcal{P}}_s = 0$, since $\sigma_t^E$ has fixed (unit) trace.

Eq. (31) has the formal solution

$$\mathcal{Q}_t \mathcal{X}_{t:s} = T_\leftarrow \exp\left[ \int_s^t dr \, \mathcal{Q}_r \mathcal{Y}_r \right] \mathcal{Q}_s + \int_s^t dr \, T_\leftarrow \exp\left[ \int_r^t dr' \, \mathcal{Q}_{r'} \mathcal{Y}_{r'} \right] \left( \mathcal{Q}_r \mathcal{Y}_r \mathcal{P}_r - \dot{\mathcal{P}}_r \right) \mathcal{X}_{r:s}, \tag{32}$$

which, when substituted into Eq. (30), gives

$$\partial_t \mathcal{P}_t \mathcal{X}_{t:s} = \left( \mathcal{P}_t \mathcal{Y}_t \mathcal{P}_t + \dot{\mathcal{P}}_t \right) \mathcal{X}_{t:s} + \mathcal{P}_t \mathcal{Y}_t \int_s^t dr \, T_\leftarrow \exp\left[ \int_r^t dr' \, \mathcal{Q}_{r'} \mathcal{Y}_{r'} \right] \left( \mathcal{Q}_r \mathcal{Y}_r \mathcal{P}_r - \dot{\mathcal{P}}_r \right) \mathcal{X}_{r:s}$$
$$+ \mathcal{P}_t \mathcal{Y}_t T_\leftarrow \exp\left[ \int_s^t dr \, \mathcal{Q}_r \mathcal{Y}_r \right] \mathcal{Q}_s. \tag{33}$$

Acting with both sides of this equation on the initial condition $\rho_{t_0}^{SE}$ and taking the partial trace withg respect to the environment, one arrives at a memory kernel master equation for $\rho_t^S = \mathrm{Tr}_E \{ \mathcal{X}_{t:t_0} \rho_{t_0}^{SE} \}$:

$$\partial_t \rho_t^S = \mathrm{Tr}_E \left\{ \mathcal{P}_t \mathcal{Y}_t \mathcal{P}_t \rho_t^S \otimes \sigma_t^E \right\}$$
$$+ \mathrm{Tr}_E \left\{ \mathcal{P}_t \mathcal{Y}_t \int_{t_0}^t dr \, T_\leftarrow \exp\left[ \int_s^t dr \, \mathcal{Q}_r \mathcal{Y}_r \right] \left( \mathcal{Q}_s \mathcal{Y}_s \mathcal{P}_s - \dot{\mathcal{P}}_s \right) \rho_s^S \otimes \sigma_s^E \right\}$$
$$+ \mathrm{Tr}_E \left\{ \mathcal{P}_t \mathcal{Y}_t T_\leftarrow \exp\left[ \int_{t_0}^t ds \, \mathcal{Q}_s \mathcal{Y}_s \right] \mathcal{Q}_{t_0} \rho_{t_0}^{SE} \right\}, \tag{34}$$

with the three terms corresponding to those in Eq. (1). For a factorising initial condition $\rho_{t_0}^{SE} = \rho_{t_0}^S \otimes \rho_{t_0}^E$, the final, inhomogeneous term vanishes for the choice $\sigma_{t_0}^E = \rho_{t_0}^E$.

Since Eq. (6) is of the form of Eq. (25), $\zeta_{\lambda,t}^S$ obeys Eq. (34) with $\mathcal{Y}_t = \mathcal{R}_{\lambda,t}$. Choosing $\sigma_t^E = \sigma^E$, $\forall t$, and hence $\mathcal{P}_t = \mathcal{P}$, $\mathcal{Q}_t = \mathcal{Q}$, $\dot{\mathcal{P}}_t = 0$, one arrives at the quantities expressed in Eqs. (9), (10), and (11).

## B  Transfer tensor derivation for generalized dynamical maps

Roughly following the derivation in Ref. [67], we can define two mappings

$$\mathbf{P}_{t_l} : \Lambda_{\lambda, t_j : t_k} \mapsto \Lambda_{\lambda, t_j : t_l} \Lambda_{\lambda, t_l : t_k}, \qquad\qquad t_j \geq t_l \geq t_k, \tag{35}$$

$$\mathbf{Q}_{t_l} : \Lambda_{\lambda, t_j : t_k} \mapsto \Lambda_{\lambda, t_j : t_k} - \mathbf{P}_{t_l} \Lambda_{\lambda, t_j : t_k}, \qquad\qquad t_j \geq t_l \geq t_k, \tag{36}$$

such that

$$\Lambda_{\lambda, t_j : t_k} = \mathbf{P}_{t_l} \Lambda_{\lambda, t_j : t_k} + \mathbf{Q}_{t_l} \Lambda_{\lambda, t_j : t_k} = \left( \mathbf{P}_{t_l} + \mathbf{Q}_{t_l} \right) \Lambda_{\lambda, t_j : t_k}, \tag{37}$$

for any $t_j \geq t_l \geq t_k$. Recursively expanding the second term of this last expression with $l = j - 1$, we arrive at

$$
\begin{aligned}
\Lambda_{\lambda, t_j : t_k} &= \left\{ \mathbf{P}_{t_{j-1}} + \mathbf{Q}_{t_{j-1}} \left( \mathbf{P}_{t_{j-2}} + \mathbf{Q}_{t_{j-2}} \right) \right\} \Lambda_{\lambda, t_j : t_k} \\
&= \left\{ \mathbf{P}_{t_{j-1}} + \mathbf{Q}_{t_{j-1}} \mathbf{P}_{t_{j-2}} + \mathbf{Q}_{t_{j-1}} \mathbf{Q}_{t_{j-2}} \mathbf{P}_{t_{j-3}} + \cdots + \mathbf{Q}_{t_{j-1}} + \cdots + \mathbf{Q}_{t_{k+1}} \mathbf{P}_{t_k} \right\} \Lambda_{\lambda, t_j : t_k} \\
&= \Lambda_{\lambda, t_j : t_{j-1}} \Lambda_{\lambda, t_{j-1} : t_k} + \left( \mathbf{Q}_{t_{j-1}} \Lambda_{\lambda, t_j : t_{j-2}} \right) \Lambda_{\lambda, t_{j-2} : t_k} + \cdots \\
&\quad + \left( \mathbf{Q}_{t_{j-1}} + \cdots + \mathbf{Q}_{t_{k+1}} \Lambda_{\lambda, t_j : t_k} \right) \Lambda_{\lambda, t_k : t_k} \\
&= \sum_{l=1}^{j-k} T_{\lambda, t_j : t_{j-l}}^{(l)} \Lambda_{\lambda, t_{j-l} : t_k},
\end{aligned}
\tag{38}
$$

where

$$
T_{\lambda, t_j : t_{j-l}}^{(l)} = \mathbf{Q}_{t_{j-1}} \cdots \mathbf{Q}_{t_{j-l+1}} \Lambda_{\lambda, t_j : t_{j-l}}
\tag{39}
$$

are the transfer tensors. These can themselves be recursively expanded, using Eq. (36), as

$$
\begin{aligned}
T_{\lambda, t_j : t_{j-l}}^{(l)} &= \mathbf{Q}_{t_{j-1}} \cdots \mathbf{Q}_{t_{j-l+2}} \Lambda_{\lambda, t_j : t_{j-l}} - \mathbf{Q}_{t_{j-1}} \cdots \mathbf{Q}_{t_{j-l+2}} \Lambda_{\lambda, t_j : t_{j-l+1}} \Lambda_{\lambda, t_{j-l+1} : t_{j-l}} \\
&= \mathbf{Q}_{t_{j-1}} \cdots \mathbf{Q}_{t_{j-l+3}} \Lambda_{\lambda, t_j : t_{j-l}} - \mathbf{Q}_{t_{j-1}} \cdots \mathbf{Q}_{t_{j-l+3}} \Lambda_{\lambda, t_j : t_{j-l+2}} \Lambda_{\lambda, t_{j-l+2} : t_{j-l}} \\
&\quad - \mathbf{Q}_{t_{j-1}} \cdots \mathbf{Q}_{t_{j-l+2}} \Lambda_{\lambda, t_j : t_{j-l+1}} \Lambda_{\lambda, t_{j-l+1} : t_{j-l}} \\
&= \Lambda_{\lambda, t_j : t_{j-l}} - \sum_{k=1}^{l-1} T_{\lambda, t_j : t_{j-k}}^{(k)} \Lambda_{\lambda, t_{j-k} : t_{j-l}},
\end{aligned}
\tag{40}
$$

with

$$
T_{\lambda, t_j : t_{j-1}}^{(1)} = \Lambda_{\lambda, t_j : t_{j-1}},
\tag{41}
$$

which is equivalent to Eq. (17) in the main text. Correspondingly, Eq. (38) is equivalent to Eq. (18).

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
